Investigation of multiple sclerosis-related pathways through the integration of genomic and proteomic data

Everest Elif 1
Ülgen Ege 2
Uygunoglu Ugur 3
Tutuncu Melih 3
Saip Sabahattin 3
Sezerman Osman Uğur 2
Siva Aksel 3
Tahir Turanli Eda eda.turanli@acibadem.edu.tr 4
1 Department of Molecular Biology and Genetics, Faculty of Science and Letters, Istanbul Technical University , Istanbul , Turkey
2 Department of Biostatistics and Medical Informatics, Faculty of Medicine, Acıbadem University , Istanbul , Turkey
3 Department of Neurology, Cerrahpaşa School of Medicine, Istanbul University-Cerrahpaşa , Istanbul , Turkey
4 Department of Molecular Biology and Genetics, Faculty of Science and Letters, Acıbadem University , Istanbul , Turkey
Sjakste Nikolajs
Electronic publication date: 2021 Sep 6
Publication date: 2021
Volume: 9
Electronic Location ID: e11922
Received 2021 Mar 17; Accepted 2021 Jul 16
Copyright: ©2021 Everest et al.
Copyright year: 2021
Copyright holder: Everest et al.
License: This is an open access article distributed under the terms of the Creative Commons Attribution License, which permits unrestricted use, distribution, reproduction and adaptation in any medium and for any purpose provided that it is properly attributed. For attribution, the original author(s), title, publication source (PeerJ) and either DOI or URL of the article must be cited.
License URL: https://creativecommons.org/licenses/by/4.0/

Keywords: Bioinformatics, Genomics, Proteomics, Multiple sclerosis

Funding: Scientific and Technological Research Council of Turkey TÜBITAK, 109S070 Istanbul Technical University, Scientific Research Projects BAP, 37426 This work was supported by the Scientific and Technological Research Council of Turkey (TÜBITAK, 109S070) for proteomic analyses and the Istanbul Technical University, Scientific Research Projects (BAP, 37426) for genomic analyses. The funders had no role in study design, data collection and analysis, decision to publish, or preparation of the manuscript.

==============================
Background

Multiple sclerosis (MS) has a complex pathophysiology, variable clinical presentation, and unpredictable prognosis; understanding the underlying mechanisms requires combinatorial approaches that warrant the integration of diverse molecular omics data.

Methods

Here, we combined genomic and proteomic data of the same individuals among a Turkish MS patient group to search for biologically important networks. We previously identified differentially-expressed proteins by cerebrospinal fluid proteome analysis of 179 MS patients and 42 non-MS controls. Among this study group, 11 unrelated MS patients and 60 independent, healthy controls were subjected to whole-genome SNP genotyping, and genome-wide associations were assessed. Pathway enrichment analyses of MS-associated SNPs and differentially-expressed proteins were conducted using the functional enrichment tool, PANOGA.

Results

Nine shared pathways were detected between the genomic and proteomic datasets after merging and clustering the enriched pathways. Complement and coagulation cascade was the most significantly associated pathway (hsa04610, P = 6.96 × 10−30). Other pathways involved in neurological or immunological mechanisms included adherens junctions (hsa04520, P = 6.64 × 10−25), pathogenic Escherichia coli infection (hsa05130, P = 9.03 × 10−14), prion diseases (hsa05020, P = 5.13 × 10−13).

Conclusion

We conclude that integrating multiple datasets of the same patients helps reducing false negative and positive results of genome-wide SNP associations and highlights the most prominent cellular players among the complex pathophysiological mechanisms.

Introduction

Multiple sclerosis (MS) is known to be an immune-mediated, neurodegenerative central nervous system (NS) disorder with complex inheritance and pathophysiological mechanisms. Although approximately common 250 genetic variants with low to modest risk effects have been associated with MS mainly by genome-wide association studies (GWAS) using rather large sample groups (International Multiple Sclerosis Genetics Consortium, 2007; Sawcer et al., 2011; Patsopoulos et al., 2017; International Multiple Sclerosis Genetics Consortium et al., 2019), known variants not only fail to explain predicted MS heritability but also cannot be efficiently translated into disease mechanisms. To date, numerous studies have also revealed potential diagnostic and prognostic biomarkers and disease-related cellular pathways that emphasize the different pathological components of the disease; however, the exact underlying mechanisms in disease development and progression are largely unknown. In order to better translate the growing number of findings into disease pathophysiology, algorithms for pathway analyses of multiple high-throughput omics data seem essential. In this context, the integration of multiple omics data is essential to better describe the complex nature of MS.

In our previous study (Avsar et al., 2015), we have conducted a cerebrospinal fluid (CSF) proteome analysis using 2D-gel electrophoresis and mass spectrophotometry and identified 151 differentially expressed proteins between an MS cohort of 179 patients with different clinical MS phenotypes and 42 non-MS controls. Later, affected Kyoto Encyclopedia of Genes and Genomes (KEGG) pathways were identified using the functional enrichment tool Pathway and Network- Oriented GWAS Analysis (PANOGA) (Bakir-Gungor, Egemen & Sezerman, 2014), revealing MS-related pathways including aldosterone-regulated sodium reabsorption pathway, renin- angiotensin system, notch signaling pathway, and vitamin digestion and absorption pathway. Here, we further explored disease-related pathways, applying single nucleotide polymorphism (SNP) genotyping on 11 MS patients, who were included in the previous proteomic study, and 60 independent, healthy individuals. Pathway enrichment analyses of the genomic and proteomic data were conducted. The two datasets were merged, highlighting the most prominent pathways that may be affected in the studied patient group.

Materials & Methods

Study groups

All patients were diagnosed at Istanbul University-Cerrahpaşa, Cerrahpaşa School of Medicine, Department of Neurology, according to the McDonald criteria (Polman et al., 2011). Eleven unrelated MS patients were randomly selected among the study group of our previous work comprising of 179 MS patients. The patient group had a heterogeneous disease presentation at the time of their CSF sample collection. Sixty independent age- and gender-matched intrinsic healthy controls were included in the analyses. All individuals in the study group are of Turkish origin. All procedures of the study were in accordance with the Helsinki Declaration of 1964 and its later amendments. The Ethics Committee of Istanbul Technical University approved the study (İTÜ-SM.İNAREK-MBG-1), and each individual of the study group gave a written informed consent form prior to the sample collection.

Genome-wide associations

DNA isolation from blood samples was performed (Roche DNA Isolation Kit for Mammalian Blood), and genotyping for 300.000 SNP markers was applied for each individual on the Illumina HumanCytoSNP-12 array. Illumina GenomeStudio software was used for quality control. All individuals showed sample call rates of more than 98% (98.65–99.63%) and were therefore included in the study. Golden Helix SNP & Variation Suite software was used for identity-by-descent detection, suggesting no relatedness between the individuals. SNP filtering was performed: SNPs on the Y chromosome, with call rates lower than 95%, minor allele frequency lower than 0.01, and in strong linkage disequilibrium (r2 > 0.5) were excluded from the study. A total of 129.547 SNPs was included in the analysis. Frequency differences of SNPs between cases and controls were assessed, and genotypic association P-values were obtained.

Pathway enrichment analysis

Genotypic associations and proteomic data were subjected to pathway enrichment analyses using the functional enrichment tool PANOGA, which reveals functionally relevant pathways by identifying genes within the pathways, incorporating protein-protein interaction (PPI) information, and extracting significant pathways. SNPs with genotypic association P-values lower than 0.05 and differentially-expressed proteins with P-values lower than 0.05 from our previous study were used for PANOGA procedure as follows: For SNP associations, because a SNP might affect more than one gene, each SNP was initially assigned to the gene on which the SNP has the most important functional effect, using the SPOT webserver (Saccone et al., 2010). Functional information of SNPs was obtained utilizing SPOT, F-SNP (Lee & Shatkay, 2007), SNPnexus (Dayem Ullah, Lemoine & Chelala, 2012), and SNPinfo (Xu & Taylor, 2009). Genes and proteins were then mapped onto a PPI network, for which Goh et al.’s human PPI network was used (Goh et al., 2007). Next, the jActive Modules algorithm (Ideker et al., 2002 was applied to identify active subnetworks containing a large number of the disease-affected genes and proteins in the PPI network. Each genotypic association and differential expression P-value was taken into account, and active subnetworks that overlap at most 50% with each other were extracted. In order to evaluate the biological importance of the subnetworks, the number of genes and proteins found in a specific KEGG pathway was compared to the total gene and protein number involved in the corresponding pathway. For this functional enrichment procedure, a two-sided hypergeometric test was used, and the Bonferroni method was applied for multiple testing corrections of the P-values. Significantly associated KEGG pathways to our patient group consisted of significantly enriched (P < 0.05) pathways for at least one of the active subnetworks. For the pathways that were enriched in multiple subnetworks, the one with the lowest P-value was reported. PANOGA was run 10 times for both genotypic associations and differentially-expressed proteins, and the lowest P-values over the 10 iterations were reported. The resulting enriched pathways for both datasets were then merged. If a given pathway was identified in both analyses, the corresponding P-values were merged using Fisher’s combined probability test. If the pathway was identified in only one of the analyses, the corresponding P-value was used as the final P-value.

Clustering of enriched pathways

Using a method previously described by Chen et al. (2014) we clustered the enriched pathways to identify similar groups and establish representative pathways. The clustering approach can be summarized as follows: initially overlap index matrix (OI) that consists of the overlap indices between all of the pairs of enriched pathways was calculated. The overlap index (OIi,j) of a pair of pathways Pi and Pj (denoting the ith and jth pathways, respectively) was defined as: OIi,j=|Gi∩Gj|minGi,|Gj|

where Gi is the set of all genes in the ith pathway. In the overlap matrix, each row (oi) represents the gene overlap profile of a pathway (i.e.,  oi is the gene overlap profile of the ith pathway). To identify similarity between each pair of pathways Pearson correlation coefficients (R) between each pair of overlap profiles (e.g.,  oi and oj) were calculated. These correlation coefficients were then converted to pairwise distances (PD) as: PD=1−R.

Using PD as the distance metric, hierarchical clustering (average-linkage) of the enriched pathways was performed. Examining the hierarchical clustering dendrogram, the dendrogram was partitioned into clusters at the manually selected PD cut-off value of 0.55. The pathway with the lowest P-value in each cluster was selected as the representative pathway for that cluster.

Results

SNP associations

All 71 individuals and a total of 129.547 SNPs passed the quality check and therefore were included in the final analyses. The Manhattan plot summarizing the genome-wide associations is given in Fig. 1. The most significantly associated SNP was rs7873, located on 3′ UTR of the IGF2 gene, whose significance level was the only value close to a classical stringent P-value cut-off for a GWAS (P = 4.39 × 10−07). When considering SNPs with a significance level of lower than a looser P-value cut off (P < 10−05) given the small sample size, 4 more SNPs have high associations with MS in the studied patient group, one of which is located on a gene: rs17187282 (P = 1.17 × 10−06), rs11688088 (P = 2.8 × 10−06), rs654188 (P = 5.24 × 10−06), and rs7092208 (P = 9.66 × 10−06, intronic variant on the MGMT gene).

Figure 1 Genome-wide Manhattan plot showing –log10 of the P-values of 129.547 SNPs (y-axis) against their genomic positions (x-axis) for associations.

The results are plotted left to right from the p-terminal ends of the chromosomes, which are shown in different colors. The blue line represents the P-value threshold (0.05).

For enrichment analysis of the SNP associations, the P-value threshold was set lower than 0.05 (Fig. 1, blue line) in order to prevent the elimination of potentially MS-associated SNPs with falsely high P-values due to the low sample size and limitations of the genome-wide association methodology. Later, elimination of the false-positive SNP associations resulted from preferring not to set a conventional stringent P-value was aimed to be achieved during the identification of MS-related pathways using SNP subnetworks, therefore ruling functionally irrelevant SNPs out. A total of 6.594 SNPs had P-values under the threshold of 0.05 (Table S1), which were included in the subsequent analyses.

MS-related pathways

The combination of enriched pathways for the resulting SNPs and differentially expressed proteins resulted in 151 enriched pathways in total (Table S2). In order to identify pathways with similar content and function, this combined list of enriched pathways was clustered (Fig. 2). The dendrogram was then manually partitioned into biologically relevant clusters, and representative pathways were established. In total, 33 clusters, therefore 33 representative pathways were obtained (Table S3). Nine of the representative pathways emerged from both genomic and proteomic analyses, among which the complement and coagulation cascade (hsa04610) was the most significantly associated pathway in the studied group (P = 6.96 × 10−30). Figure 3 shows the complement and coagulation cascades with genes identified through the genomic dataset and differentially expressed proteins identified through the proteomic dataset as a representative pathway for the relationships between the genomic and proteomic findings.

Figure 2 Clustering dendrogram of the enriched pathways.

The dendrogram shows all 151 enriched pathways, which were grouped in clusters based on their relevance to each other. The red line shows the cut-off for biologically relevant clusters, which was cut at the pairwise distance of 0.55, resulting in 33 representative clusters. Thirty-three pathways with the lowest P-values in each cluster constituted the representative pathways.

Figure 3 Complement and coagulation cascades.

The illustration of complement and coagulation cascades shows the identified genes by SNP associations (blue) and differentially expressed proteins (yellow) as a representative pathway for the merged data analysis. The pathway was adapted from KEGG: Kyoto Encyclopedia of Genes and Genomes (Kanehisa & Goto, 2000; hsa04610, 2019) and created with BioRender.com.

Regulation of actin cytoskeleton (hsa04810) and focal adhesions (hsa04510) were the two systems with the second and third lowest P-values even though there were no significant protein level changes related to these pathways (P = 9.64 × 10−27 and P = 3.29 × 10−26, respectively). Other 8 shared pathways with high significance levels are as follows: Adherens junctions (hsa04520, P = 5.38 × 10−17), colorectal cancer (hsa05210, P = 1.70 × 10−15), pathogenic Escherichia coli infection (hsa05130, P = 7.02 × 10−9), prion diseases (hsa05020, P = 2.48 × 10−5), endometrial cancer (hsa05213, P = 6.00 × 10−9), non-small cell lung cancer (hsa05223, P = 1.40 × 10−8), bladder cancer (hsa05219, P = 4.83 × 10−6), and non-homologous end-joining (hsa03450, P = 4.27 × 10−5). Table 1 shows the detected SNP associations and differentially expressed proteins in the above-mentioned pathways.

Table 1 The most prominent MS-related pathways emerged from the analyses.

KEGG terms and IDs	SNP associations	Differentially expressed proteins	FinalP-value	
	P-value	Genes	P-value		
Complement and coagulation cascades, hsa04610	1.14 × 10−16	FGA [rs2070018 ]; CR1 [rs1571344 ]; F13A1 [rs3901123; rs34736558; rs1267843; rs1267914; rs749005 ]; PLG [rs9295131 ]; F5 [rs12131397 ]; C6 [rs2921184; rs1801033 ]; PROC [rs2069933; rs1799808 ]	8.43 × 10−16	6.96 × 10−30	
Adherens junction, hsa04520	5.38 × 10−17	INSR [rs10409516 ]; LMO7 [rs9573625 ]; WASL [rs10270793 ]; EGFR [rs6970262 ]; IGF1R [rs2684788 ]; FER [rs7715933; rs7700630; rs7713591 ]; CTNNA3 [rs10997250; rs7076094; rs10997316 ]; RAC1 [rs2347338 ]; PVRL2 [rs519825 ]	2.03 × 10−10	6.64 × 10−25	
Colorectal cancer, hsa05210	1.70 × 10−15	TCF7L1 [rs6547608; rs10195517 ]; TGFB2 [rs1473527; rs1417488 ]; AXIN1 [rs3916990 ]; MAPK9 [rs6867398; rs17080136; rs11956696 ]; MAPK8 [rs17780725 ]; RAC1 [rs2347338 ]	2.88 × 10−9	2.68 × 10−22	
Pathogenic Escherichia coli infection, hsa05130	7.02 × 10−9	ROCK2 [rs6432187 ]; WASL [rs10270793 ]	3.72 × 10−7	9.03 × 10−14	
Prion diseases, hsa05020	2.48 × 10−5	NOTCH1 [rs3812605; rs3812609; rs2229974 ]; NCAM2 [rs232398]	6.31 × 10−10	5.13 × 10−13	
Endometrial cancer, hsa05213	6.00 × 10−9	TCF7L1 [rs6547608; rs10195517 ]; AXIN1 [rs3916990 ]; EGFR [rs6970262 ]; CTNNA3 [rs10997250; rs7076094; rs10997316 ]; SOS2 [rs2227276 ]	5.77 × 10−6	1.11 × 10−12	
Non-small cell lung cancer, hsa05223	1.40 × 10−8	ALK [rs7591913; rs7589120 ]; FHIT [rs643629; rs10510827; rs10510852; rs3772479; rs6778312; rs12633994; rs13317933; rs13078190 ]; SOS2 [rs2227276 ]; EGFR [ rs6970262 ]	2.82 × 10−6	1.26 × 10−12	
Bladder cancer, hsa05219	4.83 × 10−6	E2F3 [rs12527393 ]	8.47 × 10−7	1.11 × 10−10	
Non-homologous end-joining, hsa03450	4.27 × 10−5	XRCC4 [rs2662242 ]	9.88 × 10−6	9.53 × 10−9	
Top 2 pathways emerged only from SNP associations	
Regulation of actin cytoskeleton, hsa04810	9.64 × 10−27	CYFIP1 [rs7182576; rs2120968; rs17137192 ]; ROCK2 [rs6432187 ]; WASL [ rs10270793 ]; EGFR [rs6970262 ]; MYLK [rs4678060; rs3911406; rs2124508; rs702032]; GNA13 [rs12944877 ]; PAK7 [rs6118687; rs6118709 ]; RAC1 [rs2347338 ]; PAK2 [rs6583176 ]; PAK4 [rs11083505; rs692364 ]; VAV3 [rs11576720 ]; ITGA4 [rs3770111 ]; ACTN1 [rs181484 ]; ITGA2 [rs27504 ]; FN1 [rs1404772 ]; VAV2 [rs7026263 ]; TIAM1 [rs2833359; rs845972; rs9984499; rs11700792 ]; ARHGEF4 [rs2403238 ]; MYH9 [rs11703176 ]; CRK [rs6502707 ]; DOCK1 [rs10741150; rs6482855; rs10458718 ]; ARHGEF6 [rs41312580 ]; ITGA9 [rs149815; rs1984311 ]	–	9.64 × 10−27	
Focal adhesion, hsa04510	3.29 × 10−26	FIGF [rs7877192; rs6632528 ]; SHC3 [rs1331189 ]; LAMA1 [rs658121 ]; IGF1R [rs2684788 ]; MAPK9 [rs6867398; rs17080136; [rs11956696 ]; MAPK8 [rs17780725 ]; KDR [rs7692791 ]; ITGB6 [rs6730023 ]; RAC1 [rs2347338 ]; PAK4 [rs11083505; rs692364 ]; VASP [rs4803831 ]; VAV3 [rs11576720 ]; CAV3 [rs11713611 ]; ITGA4 [rs3770111 ]; ACTN1 [rs181484 ]; ITGA2 [rs27504 ]; FN1 [rs1404772 ]; MAPK10 [rs1469869 ]; ITGA10 [rs12401622 ]; ITGA8 [rs2039910; rs1451668 ]; RAPGEF1 [rs2296950 ]; TLN1 [rs2295794 ]; CRK [rs6502707 ]; DOCK1 [rs10741150; rs6482855; rs10458718 ]; SOS2 [rs2227276 ]; ITGA9 [rs149815; rs1984311 ]	–	3.29 × 10−26	

The most significantly MS-associated pathways obtained from the SNP associations and differentially expressed proteins did not overlap when the analyses were performed separately (Table S2). The merged data analysis resulted in a combined list of pathways from both datasets, revealing a different set of pathways with the lowest P-values and emphasizing the most relevant pathways that could not have been prioritized by a straight-forward, non-combinatorial omics approach. The top five pathways that emerged from the proteomic-only, genomic-only, and merged dataset analyses are given in Table 2.

Table 2 Top five pathways emerged from each dataset analysis.

KEGG Terms and IDs	P-value	
Top 5 pathways emerged from the proteomic-only data analysis	
Complement and coagulation cascades, hsa04610	8.43 × 10−16	
Adherens junction, hsa04520	2.03 × 10−10	
Prion diseases, hsa05020	6.31 × 10−10	
Colorectal cancer, hsa05210	2.88 × 10−09	
Pathogenic Escherichia coli infection, hsa05130	3.72 × 10−07	
Top 5 pathways emerged from the genomic-only data analysis	
Regulation of actin cytoskeleton, hsa04810	9.64 × 10−27	
Focal adhesion, hsa04510	3.29 × 10−26	
ErbB signaling pathway, hsa04012	1.28 × 10−23	
Rap1 signaling pathway, hsa04015	2.35 × 10−20	
Axon guidance, hsa04360	6.36 × 10−20	
Top 5 pathways emerged from the merged data analysis	
Complement and coagulation cascades, hsa04610	6.96 × 10−30	
Regulation of actin cytoskeleton, hsa04810	9.64 × 10−27	
Focal adhesion, hsa04510	3.29 × 10−26	
Adherens junction, hsa04520	6.64 × 10−25	
ErbB signaling pathway, hsa04012	1.28 × 10−23	

Discussion

The complement and coagulation cascade (CCC) pathway emerged as the most significantly associated pathway to MS in our patient group, with many SNP associations on different genes and differentially expressed CSF proteins as shown in Fig. 3. The complement and the coagulation systems are two closely linked cascades, both having roles in immunity. In a study by Magliozzi et al. (2019), CSF proteomic profiles of MS patients with low or high cortical lesion load were compared, revealing that the identified differentially expressed proteins were mainly involved in the complement and coagulation cascade. Examination of white matter lesions of MS patients has revealed dysregulation of coagulation-associated proteins in chronic active plaques involving Serpin A5 and tissue factor (Han et al., 2008). Also, significantly upregulated Serpin E1, another CCC component, was reported in the post-mortem cortex of progressive MS patients (Yates et al., 2017). In our study, even though the genes and differentially expressed proteins involved in CCC do not overlap, the pathway was found 10 times for each dataset in PANOGA, resulting in a high association with MS. Our results confirm the previous findings suggesting the involvement of CCC alterations in MS pathophysiology and highlight other pathway components that may also be responsible for these alterations.

Disruption of blood–brain barrier (BBB) integrity is one of the hallmarks in MS pathophysiology, during which massive leukocyte infiltration occurs across the damaged BBB into the CNS (Alvarez, Cayrol & Prat, 2011) Tight and adherens junctions between the endothelial cells of BBB have significant importance for normal immune surveillance in the CNS. Tight junctions consist of claudins, occludin, junctional adhesion molecules, and Zonula Occludens (Hawkins & Davis, 2005), whose dysfunctions leading to BBB abnormalities in MS have been previously reported (Kirk et al., 2003; Padden et al., 2007; Plumb et al., 2002). Cadherins form adherens junctions via homophilic interactions between endothelial cells and bind cytoskeletal components through cytoplasmic catenin proteins (Dejana, Orsenigo & Lampugnani, 2008). Although adherens junctions are required for the overall junctional organization to maintain the BBB integrity, their roles in normal physiology and diseased states have not been well-established. In a previous study, the level of an adherens junction protein, β-catenin, was similar in progressive MS and non-MS brain sections (Padden et al., 2007). In our patient group, adherens junctions, which were not detected in our previous proteomic study, emerged as the second most significantly altered cellular components with SNP associations on different genes and differentially expressed proteins indirectly related to the pathway. Our findings indicate that adherens junctions may have more influence on BBB function than it has been thought and are needed to be explored in both normal physiology and MS pathophysiology in more detail.

Focal adhesions and actin cytoskeleton regulation are two interrelated pathways that emerged with low final P-values from the analyses, even though no significant protein expression changes directly or indirectly related to these systems were detected. Organized clusters of focal adhesion complexes connect extracellular matrix through transmembrane integrin proteins to the intracellular actin cytoskeleton through other focal adhesion proteins in the complexes and mainly have roles in cell motility (Wozniak et al., 2004). During leukocyte infiltration across the BBB, α4-integrin proteins on leukocytes form firm connections with the endothelial surface, initiating the process (Berlin et al., 1995; Vajkoczy, Laschinger & Engelhardt, 2001). In this context, Natalizumab (Antegren, Elan Pharmaceuticals and Biogen), a monoclonal antibody against α4-integrin, has been used in MS treatment, which suppresses inflammatory activity by inhibiting leukocyte migration to the inflammation areas (Polman et al., 2006). We detected many SNP associations on genes encoding for many focal adhesion proteins, including the α4-integrin-coding ITGA4 gene. Focal adhesion molecules and their combinations to create different complexes are diverse, and regulation differences of these interactions for specific cell behaviors are yet to be elucidated.

In the studied MS patients, increased CSF nucleolin expression and SNP associations on WASL and ROCK2 genes were detected, all of which are components of the Pathogenic Escherichia coli infection pathway. Neural Wiskott-Aldrich syndrome protein encoded by WASL and Rho-associated protein kinase 2 encoded by ROCK2 are both regulators of the actin cytoskeleton (Miki, Miura & Takenawa, 1996; Gallo et al., 2012), and the involvement of the pathway may indicate induction of inflammatory cell mobility. Prion diseases are fatal conditions presenting neuronal degeneration, and the emergence of the pathway from our analyses emphasizes some shared molecular alterations with MS pathophysiology, as both neurodegeneration and neuroinflammation also occur in prion diseases (Perry, Cunningham & Boche, 2002; Eikelenboom et al., 2002). We have also detected a number of cancer pathways involving both SNP associations and differential CSF expressions. Cancer risk among MS patients has been investigated in a number of studies with diverse results, revealing unchanged (Nielsen et al., 2006), reduced (Bahmanyar et al., 2009), and increased (Grytten et al., 2020) cancer rates. Altered cancer pathways in our study group may be attributed to changes in immunological mechanisms involving the same components as in mechanisms against anti-tumor surveillance. Other pathways that emerged only from the SNP associations included a number of immunological and neurological mechanisms. T cell receptor signaling pathway (hsa04660, P = 1.86 × 10−19), chemokine signaling pathway (hsa04062, P = 4.38 × 10−19), and Fc-gamma R-mediated phagocytosis (hsa04666, P = 8.78 × 10−16) were the immune system-related pathways. Nervous system-related pathways included axon guidance (hsa04360, P = 6.36 × 10−20), retrograde endocannabinoid signaling (hsa04723, P = 6.74 × 10−18), neurotrophin signaling pathway (hsa04722, P = 8.03 × 10−18), glutamatergic synapse (hsa04724, P = 1.74 × 10−15), cholinergic synapse (hsa04725, P = 7.79 × 10−15), and dopaminergic synapse (hsa04728, P = 5.83 × 10−12). Also, the Rap1 signaling pathway (hsa04015, P = 2.35 × 10−20) was detected, further supporting the involvement of adherens junction, focal adhesion, and regulation of actin cytoskeleton pathways.

Working on small sample sizes for genome-wide associations leads to enormous numbers of false-positive SNP associations. Using large sample sizes, on the other hand, may result in false-negative results due to the low effect sizes of disease-associated SNPs. One limitation of our study is the considerably low sample size in the assessment of SNP associations. In this regard, we aimed to take advantage of having both DNA and CSF samples of the same affected individuals. Therefore, we aimed to include both false-positive SNP associations and prevent missing out SNPs with falsely high P-values in the first place by setting a non-stringent P-value cut-off. We then aimed to exclude false-positive associations during the pathway enrichment analyses in which interacting SNP subnetworks were used to identify the pathways, eliminating functionally irrelevant SNPs. Integrative analyses of the genomic and proteomic datasets determined the shared pathways between the two datasets, which seem to represent the pathophysiological mechanisms. Another limitation of the study is that we have used a genotyping chip with a relatively low SNP number (300K) since it was one of the most powerful chips when the genotyping had been carried out. Lastly, we had not obtained mRNA samples of the study participants at the time of their sample collection, which would lead to a more comprehensive analysis of the altered pathways in the studied patient group.

Conclusions

Here, we provide an integrated pathway analysis of whole-genome SNP and CSF proteome datasets obtained from the same MS patients, highlighting the most prominent molecular pathways that may be altered in the studied patient group. We confirmed the involvement of some of the previously known players in MS pathophysiology and suggested possible roles of some others. We conclude that this approach provides a better placing of separate genomic and proteomic datasets into a biological context and is useful for elucidating disease mechanisms.

Supplemental Information

Supplemental Information 1 SNPs included in the pathway analysis

Supplemental Information 2 All pathways found by the combined pathway enrichment analysis of the SNPs and proteins

Supplemental Information 3 Representative pathways found by clustering of the merged enriched pathways

Supplemental Information 4 Raw genotype data obtained using HumanCytoSNP-12 (Illumina)

We thank the patients and volunteers who agreed to contribute to the study.

Additional Information and Declarations

Competing Interests

Author Contributions

Human Ethics

Data Availability

The authors declare there are no competing interests.

Elif Everest performed the experiments, analyzed the data, prepared figures and/or tables, authored or reviewed drafts of the paper, and approved the final draft.

Ege Ülgen analyzed the data, prepared figures and/or tables, authored or reviewed drafts of the paper, and approved the final draft.

Ugur Uygunoglu, Melih Tutuncu, Sabahattin Saip and Aksel Siva conceived and designed the experiments, authored or reviewed drafts of the paper, diagnosed the patients, and approved the final draft.

Osman Uğur Sezerman and Eda Tahir Turanli conceived and designed the experiments, authored or reviewed drafts of the paper, and approved the final draft.

The following information was supplied relating to ethical approvals (i.e., approving body and any reference numbers):

The Ethics Committee of Istanbul Technical University approved the study (İTÜ-SM.İNAREK-MBG-1).

The following information was supplied regarding data availability:

The raw data are available in the Supplemental Files.

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
