# Peer review of "Investigation of multiple sclerosis-related pathways through the integration of genomic and proteomic data"

_PeerJ, doi:10.7717/peerj.11922_

## Round 0.1 · original submission · Minor Revisions

Please, make change suggested by the Reviewers.

Reviewer 1 ·

Basic reporting

The authors used combinational omics approaches to reveal functional pathways of Multiple sclerosis and such an approach is demanded in the field. Nevertheless, the main message of the work should be more widely disclosed.
English - suitable.
Cited references - suitable.
Fig. 1 needs to be improved. No blue threshold line indicated.

Experimental design

Methods part is well described; investigation performed according to the ethical standards. I missed the visualization of the main findings, where the protein level and SNP data would be shown in a single figure. The possible association between SNP and protein expression is described in the text, nevertheless, the additional figure would let to better reveal the association and translate the main message of the analysis. Since narrowing down analysis enabled the selection of the most important targets possibly associated with MS, these can be deeper analyzed revealing protein level and SNP link.

The authors are comparing two distal biological elements - initial: DNA code changes (SNPs) and final - protein level, nevertheless, there are numbers of intermediate regulation between these elements. It would be very beneficial if authors could analyze for example mRNA level of selected targets in selected samples. If it is not possible, the possible impact should be considered in the discussion part.

Validity of the findings

Since the idea to compare different biological elements data (SNPs, and protein expr.) itself is great, the result should be treated with cautions, since there are many intermediate biological elements, which are not measured. Such design might generate false-positive associations between SNP-proteil level- and MS.

Additional comments

Comments are given in the section above.
Why you selected only 11 patients for the analysis since you have a bigger cohort?
Are you the first who analyse protein level and SNP association in MS?
Why only a qualitative comparison of MS and molecular features were made, does the "disease progress" for all the patients was similar? It would be worth including the clinical characteristics of patients when comparing molecular data as well.

Reviewer 2 ·

Basic reporting

One main suggestion from my side would be to broaden the Introduction. The discussion part sufficiently covers all the questions raised during the given study, but I would suggest moving the first paragraph of the Discussion (L193 – L201) to the Introduction section.

Experimental design

The experimental part is flawlessly designed, and the Materials and Methods section covers all the necessary information. The ethical aspect is described in the Study group subsection.

Validity of the findings

The current pool of MS genetic data is relatively small. So far, there is little evidence to correlate MS genetic data with the clinical phenotype. But recent studies confirm that MS genetics may indeed describe the severity of the disease. The authors did a great job attempting to fill in the knowledge gap within the genetic aspect of MS.

Additional comments

General suggestions:
L2 – “the” before “integration”
L46 and L80 – comma after “independent”
L81 – split into two sentences. “…were conducted. The two datasets…”
L126 – perhaps “corrections” instead of “correction”
L132 – change “analyses” to “studies”
L157 – comma after “rs7873”
L158 – cut-off
L202 – The complement
L211 – CCC component
L228 – L229 remove “found to be”
L236 – re-phrase as “actin cytoskeleton regulation”
L244 – comma instead of semicolon
L258 – comma instead of semicolon
L259 – also occur
L278 – dot after “associations” and remove “whilst”
L280 and through the text – remove “very”. Consider not using “very” at all or replace with “rather, relatively, considerably…”
L281 – L283 consider rewriting as “Therefore, we aimed to include both false-positive SNP associations and prevent missing out…”
L287 – change “be effective for representing” to represent
L290 – change “at the time” to “when”
L301 – “agreed” instead of “accepted”

Reviewer 3 ·

Basic reporting

Thw paper by Everest et al. presents a pathway enrichment analysis on merged genotyping and proteomic data.
The paper is clearly writter and clearly presented.

Experimental design

The experimental design involve the analysis of a previously released proteomic dataset and of a new genotype dataset.

Enriched pathways obtained by the merging of the two datasets are presented and contrasted with those obtained from the anaysis of the proteomic dataset. The comparison should be better shown and it would be interesting to compare with results obtained using genotype data only .

The procedure of analysis, particularly for genotype data, involves the adoption of many external resources (e.g. SPOT, human PPI network) and some of them seems a bit outdated. Are they critical for the analysis and the drawn conclusions? In particular, "the gene on which the SNP has the most important functional effect" is a quite general sentence for a very complex association and it deserves more comment, in my opinion.

If possible, results could be corroborated by adopting alternative methods (but I undestand this can be time consuming and not essential in the economy of the paper)

Validity of the findings

I cannot comment the medical and clinical significance of the findings. On the point of view of the analysis, I think that the statement " We conclude that integrating multiple datasets of the same patients helps reducing false negative and positive results of genome-wide SNP associations and highlights the most prominent cellular players among the complex pathophysiological mechanisms" would be strengthened by a more thorough comparison of the pathways enriched with proteomics and genotyping data, separately, and with the merged dataset

---

## Round 0.2 · accepted · Accept

Congratulations, your paper is now accepted!

Reviewer 2 ·

Basic reporting

No comment

Experimental design

No comment

Validity of the findings

No comment

Additional comments

I am fully satisfied with the revision. All suggestions were taken into account. No further comments from my side.

Reviewer 3 ·

Basic reporting

The current version of the paper basically addresses all previously raised concerns

Experimental design

The only point possibily deserving deeper (although optional) discussion is the fact that a quite outdated protein-protein interaction network is adopted, while more than 10 years later the picture is probably changed.

Validity of the findings

No further comment